# Mediterranean Shrub Species as a Source of Biomolecules against Neurodegenerative Diseases

**DOI:** 10.3390/molecules28248133

**Published:** 2023-12-16

**Authors:** Natividad Chaves, Laura Nogales, Ismael Montero-Fernández, José Blanco-Salas, Juan Carlos Alías

**Affiliations:** Department of Plant Biology, Ecology and Earth Sciences, Faculty of Science, Universidad de Extremadura, 06080 Badajoz, Spain; lnogalesg@unex.es (L.N.); ismonterof@unex.es (I.M.-F.); blanco_salas@unex.es (J.B.-S.); jalias@unex.es (J.C.A.)

**Keywords:** natural antioxidants, neuroprotective compounds, phenols, Mediterranean species

## Abstract

Neurodegenerative diseases are associated with oxidative stress, due to an imbalance in the oxidation-reduction reactions at the cellular level. Various treatments are available to treat these diseases, although they often do not cure them and have many adverse effects. Therefore, it is necessary to find complementary and/or alternative drugs that replace current treatments with fewer side effects. It has been demonstrated that natural products derived from plants, specifically phenolic compounds, have a great capacity to suppress oxidative stress and neutralize free radicals thus, they may be used as alternative alternative pharmacological treatments for pathological conditions associated with an increase in oxidative stress. The plant species that dominate the Mediterranean ecosystems are characterized by having a wide variety of phenolic compound content. Therefore, these species might be important sources of neuroprotective biomolecules. To evaluate this potential, 24 typical plant species of the Mediterranean ecosystems were selected, identifying the most important compounds present in them. This set of plant species provides a total of 403 different compounds. Of these compounds, 35.7% are phenolic acids and 55.6% are flavonoids. The most relevant of these compounds are gallic, vanillic, caffeic, chlorogenic, *p*-coumaric, and ferulic acids, apigenin, kaempferol, myricitrin, quercetin, isoquercetin, quercetrin, rutin, catechin and epicatechin, which are widely distributed among the analyzed plant species (in over 10 species) and which have been involved in the literature in the prevention of different neurodegenerative pathologies. It is also important to mention that three of these plant species, *Pistacea lentiscus*, *Lavandula stoechas* and *Thymus vulgaris*, have most of the described compounds with protective properties against neurodegenerative diseases. The present work shows that the plant species that dominate the studied geographic area can provide an important source of phenolic compounds for the pharmacological and biotechnological industry to prepare extracts or isolated compounds for therapy against neurodegenerative diseases.

## 1. Introduction

### 1.1. Brief Description of Neurodegenerative Diseases and Their Causes

Neurodegenerative diseases, diabetes, cardiovascular diseases, sarcopenia, and cancer are associated with the “free radical theory” of aging [1,2,3]. This theory is based on the structural damage-based hypothesis claiming that tissue dysfunction due to aging can be attributed to the accumulation of oxidative damage of macromolecules by free radicals [1]. Oxidative stress results from an imbalance in reduction and oxidation reactions at the cellular level. The consequence of this imbalance is the formation of reactive oxygen or nitrogen species (ROS/RNS) and sometimes it can be attributed to a decrease in the level of antioxidant defense [4,5]. In particular, the excessive production of ROS contributes to oxidative stress leading to neuronal cell death and an alteration of brain function [2,6]. The central nervous system is vulnerable to oxidative stress since it has a large requirement for oxygen and has a lower amount of antioxidant enzymes, compared with other tissues [7,8].

Harman et al. [1] extended the “free radical theory” of aging to the “mitochondrial theory of aging”, which states that ROS accumulation induces mitochondrial dysfunction, which contributes to aging and the development of related diseases [9,10,11]. Over the last decade, a connection between mitochondria and longevity has become increasingly evident. Mitochondria are also regulators of some types of cell death, such as apoptosis, and thus, their mitochondrial dysfunction might affect the lifespan of individuals presenting defects in this organelle [11].

Neurodegenerative diseases are frequently associated with neuroinflammation, which is a process related to oxidative and nitrosative stress. The inflammatory response is further propagated by the activation of glial cells and the modulation of constitutively expressed extracellular matrix proteins [12,13]. 

Many neurodegenerative diseases have been described to be highly prevalent in the population and have a high socioeconomic impact. Alzheimer’s disease (AD), Parkinson’s disease (PD), Huntington’s disease (HD), amyotrophic lateral sclerosis, and frontotemporal dementia are some examples [14,15]. All these diseases, specifically AD and PD, are associated with high morbidity and mortality and represent a primary health problem, especially in the aged population [16]. These disorders share common pathological characteristics, such as the induction of oxidative stress, abnormal protein aggregation, perturbed Ca^2+^ homeostasis, excitotoxicity, inflammation and apoptosis [17,18].

AD is a progressive neurological condition and the world’s most common form of dementia [19]. The pathological characteristics include extracellular deposits of amyloid β, (Aβ), intracellular formation of neurofibrillary tangles and loss of neuronal synapses and pyramidal neurons [20]. Aβ deposits derive from amyloid precursor protein and the neurofibrillary tangles containing an abnormally phosphorylated form of tau, which is a microtubule-associated protein [21]. A growing body of research supports that Aβ aggregation and decreased Aβ clearance are the leading causes of this disease onset.

Different studies indicate that oxidative stress plays a fundamental role in the development and evolution of AD. For example, elevated ROS production has been shown to initiate toxic amyloid beta precursor protein processing, thereby triggering Aβ generation [22]. These ROS are primarily generated via NADPH oxidase 2, which is well associated with inflammation and amyloid plaque deposition, leading to mitochondrial dysfunction and decreased glutathione levels. Neurons contain a high amount of polyunsaturated fatty acids that can interact with ROS, leading to a self-propagating cascade of lipid peroxidation and molecular destruction [23]. Products of lipid peroxidation have also been shown to be elevated in blood samples and brains of AD patients at autopsy [24]. It has also been correlated with the initial stage of the disease DNA oxidative damage in the AD brains, due to increased expression of ERCC-80 and 89 genes related to DNA repair activity [25].

On the other hand, the “cholinergic hypothesis of AD” is based on acetylcholine deficiency [26,27]. This neurotransmitter is involved in cognition and memory processes that are known to be decreased in AD. Thus, cholinesterase inhibitors are the first line of therapy for the management of AD [28,29].

PD is the second most common age-related neurodegenerative disorder in the central nervous system. This disease is a clinical syndrome characterized by motor impairments, including bradykinesia, resting tremor, muscle rigidity, loss of postural reflexes, freezing phenomenon and flexed posture. PD involves the loss of dopaminergic neurons of the pars compacta region of the substantia nigra and the accumulation of intracellular proteins (synucleins), leading to cognitive and motor deterioration in people who suffer from it [30,31,32,33]. 

It is possible that processes, such as oxidation, may be responsible for the gradual dysfunction that can be manifested throughout the disease. Previous publications have reported evidence of this oxidative stress through the detection of oxidized DNA, lipids, and proteins in the brain tissues of PD patients [34]. Dopaminergic neurons contain large amounts of ROS derived from dopamine’s enzymatic and non-enzymatic metabolism. Dopamine may be catabolized by monoamine oxidase (MAO) in a process that generates 3,4-dihydroxyphenyl-acetaldehyde, ammonia and H_2_O_2_, which reacts with Fe^2+^ to form hydroxyl radical. In addition, dopamine oxidation may spontaneously generate 6-hydroxydopamine, which is subsequently transformed into reactive electrophilic molecules in the presence of oxygen [35]. On the other hand, several studies suggest that the overexpression or misfolding of α-synuclein increases ROS production and cell susceptibility to oxidative stress [36,37].

In general, the treatments for neurodegenerative diseases tend to be limited in their therapeutic approach, due to their symptom management but non-curative nature [38], and the continuous use of certain conventional drugs generates many adverse effects, such as nausea, diarrhea, eating disorders and kidney and liver affectations [39]. Therefore, it is necessary to find complementary and/or alternative treatments. Several clinical trials have proved the implication of natural products as antioxidant agents (e.g., ferulic and *p*-coumaric acids, resveratrol, catechin, epi-catechin, quercetin, ginsenosides.) [40,41,42,43] and, given the role of oxidative stress in the pathogenesis of neurodegenerative diseases, these compounds can be a good therapeutic alternative against these diseases.

### 1.2. Phenolic Compounds: Their Importance and Implication in Neurodegenerative Diseases 

Many studies have focused on natural phytocomponents as important bioactive molecules against aging-related chronic diseases, including neurodegenerative diseases [44,45,46]. The wide and countless number of natural compounds from plants, animals, fungi and microorganisms provide a rich and unique source for new drug search [47], with plants being the main source of these compounds. In most cases, the biological activity attributed to plant extracts derives from secondary metabolites, which include two extensive categories: nitrogen-containing and non-nitrogen-containing compounds [48,49]. In the latter category, phenols are one of the most extensive groups of secondary metabolites in the plant kingdom [50]. Structurally, they are characterized by the presence of at least one hydroxyl functional group (-OH) linked to an aromatic ring [51]. Polyphenol classification is based on the number of phenol rings in the molecule, and the main subgroups include phenolic acids, coumarins, stilbenes, flavonoids, tannins and lignans [50]. These compounds exert different biological activities, including antioxidant, antiallergic, anti-inflammatory, antiviral, antiproliferative and anticarcinogenic effects [52,53].

One of the most remarkable functions of phenols is their capacity to suppress oxidative stress and neutralize free radicals. They can act as reducing agents, metal chelators, free-radical scavengers, enzyme modulators and regulators of diverse proteins and transcriptional factors (Figure 1) [54,55]. The antioxidant potential of these compounds confers therapeutic activities for a wide variety of diseases, such as cardiovascular diseases, cancer, liver diseases, diabetes and neurodegenerative disorders [56,57].

It has been demonstrated that phenols can inhibit the aggregation of proteins involved in various neurodegenerative pathologies in which cognitive deterioration occurs, including AD, PD, dementia with Lewy bodies, and multiple system atrophy [58]. In fact, studies conducted with flavonoids show that these compounds would be involved in preventing neurodegeneration [59]. Their bioactivity is attributed to their antioxidant effect and their capacity to inhibit acetylcholinesterase (AChE)/butyrylcholinesterase (BChE) [19] and the GABA receptor [60], alleviate mitochondrial dysfunction [61], modulate neuronal signaling pathways critical for the control of neuronal resistance to neurotoxic oxidants, inhibition inflammatory mediators [62] and chelation of transition metal ions [59].

It has been shown that the interaction of flavonoids with these receptors depends on the structure [63], implying that not all phenols have the same activity and importance as agents to prevent or treat neurodegenerative diseases. It has been suggested that B-ring hydroxylation is a differentiating element in the action exerted by flavonoids, particularly the positive contribution of 5-dehydroxylation and 3′,4′-ortho-dihydroxylation on the B-ring [64]. Furthermore, a study on flavonoid-PI3-kinase interaction has further confirmed the pivotal role of B-ring hydroxylation [65], and highly sensitive allosteric modulation has been proposed [60].

Other studies have reported the direct involvement of phenolic compounds in preventing various pathologies associated with oxidative stress. One of these compounds is resveratrol, a phenol that can directly target multiple signaling cascades involved in neurodegenerative diseases, such as anti-inflammatory activity and inhibiting the aggregation of toxic Aβ amyloid protein [66]. Another example is 3,4,5–trihydroxybenzoic acid, a phenol that inhibits the plasma membrane Pdr5p efflux pump in AD124567 yeast strain overexpressing the PDR5 gene [67]. Another study demonstrated that 4-hydroxy-3-methoxybenzaldehyde represses translation in yeast, as concluded by the accumulation of processing bodies and stress granules composed of non-translating mRNAs and proteins after 4-hydroxy-3-methoxybenzaldehyde exposure [68].

In studies conducted in *Saccharomyces*, Sunthonkun et al. [69] observed the positive effects of 3,4-dihydroxybenzoic acid against aging. In this sense, 3,4-dihydroxybenzoic acid positively modulated the life span of *Saccharomyces* by reducing ROS, conferring cells greater resistance against free radicals. According to these authors, regarding the reduction of ROS levels, 3,4-dihydroxybenzoic acid seems to imitate the effect of the inactivation of proteins such as Sir2 (silent information regulator 2), Tor1 (protein kinase) or Sch9 (protein kinase).

Considering all this information and the multifactor origin of neurodegenerative disorders, it is interesting and necessary to delve into the study of natural multitarget compounds and their bioavailability [70,71].

Mediterranean ecosystems show a great diversity of plant species derived from the specific climatic conditions and the heterogeneity of their habitats [72]. The species that dominate these habitats endure harsh conditions due to the frequency of wild fires, high temperatures, water stress in summer and herbivory [73]. These unfavorable conditions stimulate the production of compounds derived from secondary metabolism, specifically phenolic compounds [74,75], which play an important ecological role in the adaptive response to these unfavorable conditions. Therefore, the species of these ecosystems may constitute an important and diverse source of compounds that should be studied.

This work aimed to evaluate the potential of Mediterranean shrub species as a source of phenolic compounds. To this end, we selected the shrub species that dominate a particular geographic area of the Iberian Peninsula.

## 2. Description of the Study Area and Article Search Strategy

The study area selected was Extremadura, a region of the Western Iberian Peninsula with a surface of 41,635 km^2^. From a biogeographic perspective, it is in the Mediterranean region and is characterized by a diverse set of plant associations that result from the interaction of its biotic and abiotic factors. The bioclimatic floors and levels that may be found in the region of Extremadura are mesomediterranean, supramediterranean and orosubmediterranean [76], and they are associated with a rainfall of 200–2000 mm/year and an average annual temperature of 4–19 °C. These conditions are responsible for the wide variability of the plant landscape of this region, which is represented by the vegetation sets described in “Plant Landscape and Dynamics in Extremadura” [77], with the following dominating shrub formations: orophilous laburnum and creeping juniper, heath and rock rose, broom and rotem, thyme and cantuesar, gorse and basophilic rock rose, wild olive and mastic, strawberry tree, arborescent juniper, kermes oaks, and garrigues and thorny bushes (brambles and thorns). These groupings are characterized and dominated by the following 24 species: *Cistus ladanifer*, *Cistus salviifolius*, *Cistus monspeliensis*, *Cistus crispus*, *Cistus albidus*, *Cistus populifoius*, *Cytisus multiflorus*, *Cytisus scoparius, Cytisus striatus*, *Erica multiflora*, *Erica scoparia*, *Erica australis*, *Calluna vulgaris*, *Myrtus communis*, *Pistacea lentiscus*, *Pistacea terebinthus*, *Rosmarinus officinalis*, *Quercus ilex*, *Quercus suber*, *Arbutus unedo*, *Lavandula stoechas*, *Thymus mastichina*, *Thymus vulgaris* and *Rubus ulmifolius*.

In this study, we reviewed the works conducted on the identification of phenolic compounds in the 24 selected species. The data reflect the compounds identified in studies published between 1996 and 2022 in the Pubmed, ScienceDirect and Scopus databases. We selected the articles where the identification of these compounds was supported, at least, by techniques such as high-performance liquid chromatography (HPLC)–photodiode array detection (DAD)–mass spectrometry (MS), which provide reliable information about the constitution of phenolic compounds. Articles that were not available in full-text were not considered. Moreover, articles without a clear experimental procedure were also excluded.

## 3. Description and Classification of the Phenolic Compounds Present in the Selected Species

Table 1 presents the list of phenolic compounds that have been identified by different authors in the 24 species selected for this study. A total of 403 different phenolic compounds can be found in this entire set of species. These compounds belong to different classes or groups: phenolic acids, flavonoids, other polyphenols, lignans and stilbenes. Each of these groups accounts for 35.7%, 55.6%, 5.7%, 2% and 0.7%, respectively (Table 2). As can be observed, the largest group is constituted by flavonoids; thus, these species are an important and diverse source of these phenols.

**Table 1 molecules-28-08133-t001:** Phenolic compounds identified in the 24 species selected in this study.

	Species																							
	Cl	Cs	Cm	Cc	Ca	Cp	Cym	Cys	Cyst	Em	Es	Ea	Cv	Mc	Pl	Pt	Ro	Qi	Qs	Au	Ls	Tm	Tv	Ru
Class: Phenolic acid. Sub Class: Hydroxybenzoic acids																								
Acetovanillone								+																
Anisic acid	+																+			+				
Benzoic acid	+																							
Benzyl benzoate	+																							
Methyl benzoate	+																							
Castalagin																				+				
Cornusilin	+	+				+																		
3-hydroxybenzoic acid																	+				+		+	
4-hydroxybenzoic acid	+														+		+			+	+		+	+
4-hydroxybenzoic acid glucuronide																				+				
Glucose p-hydroxy benzoate																							+	
Dihydroxy-methoxybenzoic acid																+					+			
Dihydroxybenzoic acid di-pentoside																					+			
Dihydroxybenzoic acid hexoside																				+				
Ducheside A	+																							
3,4′-dihydroxypropiofenone-3-glucoside	+	+	+																					
3-*O*-galloylquinic acid (Theogallin)		+													+					+				
3-*O*-galloylshikimic acid																				+				
3,4-Di-*O*-galloylquinic acid		+													+	+				+				
5-*O*-galloylquinic acid														+						+				
5-*O*-galloylshikimic acid																				+				
Galloyl arbutin																				+				
Galloyl glucose	+	+	+		+																			
Galloyl glucuronide																				+				
Galloyl hexoside																+				+				
Galloyl-bis-HHDP glucose																								+
Galloyl-HHDP-DHHDP-hexoside																				+				
Galloyl-HHDP-hexoside																				+				
Digallic acid																+								
Digalloyl glucose	+																			+				
Digalloyl shikimic acid																				+				
Digalloyl-HHDP-hexoside																				+				
Digalloylarbutin																				+				
Digalloylquinic shikimic acid																				+				
Tetra-galloyl-hexoside																				+				
Trigalloylquinic acid		+													+					+				
Trigalloylshikimic acid																				+				
Pentagalloyl glucose							+																	
Ellagic acid	+						+					+			+			+		+	+			+
Ellagic acid-7-xyloside	+	+			+	+																		
Ellagic acid arabinoside																				+				
Ellagic acid diglucoside																				+				
Ellagic acid glucoside																				+				
Ellagic acid glucuronide																								+
Ellagic acid hexoside																				+				
Ellagic acid mannopyranoside																								+
Ellagic acid pentoside																				+				+
Ellagic acid xylofuranoside																								+
Ellagitannin																				+				
Methylellagic acid rhamnoside																				+				
3,3′-di-*O*-Methylellagic acid 4-*O*-β-D-(2″-acetyl) glucoside			+																					
Gallic acid (3,4,5-trihydroxybenzoic acid)	+	+	+	+	+	+	+	+							+	+	+	+	+	+	+		+	+
Gallic acid dihexoside																				+				
Gallic acid glucoside																				+				
Gallic acid hexoside																				+				
Gallotannin																				+				
Ethyl gallate																				+				
Methyl gallate																		+						
Gentisic acid																	+				+		+	
Gentisoyl glucoside	+		+			+																		
Gentisoyl hexoside	+																							
Glucogallin	+				+	+									+									
Hexahydroxydiphenoyl-glucose	+	+	+	+	+	+																		
Isoeugenol																	+							
Lambertianin C																								+
Methoxysalicylic acid																						+		
Protocatechualdehyde (3,4-dihidroxy-benzaldehyde)								+																
Protocatechuic acid (3,4-dihydroxy-benzoic acid)								+							+	+	+	+		+	+		+	+
Protocatechuic acid glucoside																				+			+	
Punicalagin	+	+	+			+																		
Punicalin	+	+	+			+																		
Punicalagin-gallate	+	+				+																		
Sanguiin H-10																								+
Sanguiin H-10 isomer																								+
Shikimic acid gallate																				+				
Shikimic acid dimer		+	+		+																			
Strictinin ellagitannin																				+				
Syringic acid							+	+							+	+	+			+			+	+
Syringyl-shikimic acid		+	+		+																			
TriGG-dehydrohexahydroxydiphenoyl (DHHDP)-glucose																				+				
Uralenneoside																+								
Vanillic acid	+						+					+			+		+	+		+	+		+	+
Vanillic acid sulfoquinovoside																					+			
Class: Phenolic acid. Sub Class: Hydroxycinnamic acids																								
Caffeic acid (3,4-dihydroxycinnamic acid)		+					+	+							+	+	+	+		+	+	+	+	+
caffeic acid 4-*O*-glucoside																							+	
Caffeic acid derivate												+								+	+		+	
Caffeic acid hexoside																							+	+
Caffeic acid trimer																							+	
Dihydrocaffeic acid																					+		+	
Caffeoyl arbutin																				+				
Caffeoyl ferulic acid		+																						
Caffeoyl feruloyl tartaric acid																					+			
Caffeoyl hexoside																					+			
Caffeoyl hexoside derivative																					+			
4-*O*-Caffeoyl quinic acid										+							+							+
Caffeoyl quinic acid glucoside		+	+		+																			
Caffeoyl quinic acid derivative		+																					+	
Caffeoyl tartaric acid (caftaric acid)					+		+														+			
Dicaffeoyl shikimic acid		+																						
1,4-dicaffeoyl quinic acid															+									+
3,5-dicaffeoyl quinic acid																								+
6-Caffeoyl sucrose																					+			
Chlorogenic acid (3-*O*-caffeoylquinic acid)		+		+	+		+	+		+			+		+					+	+	+	+	+
Methyl caffeate																					+			
Cinnamic acid	+														+	+	+			+	+		+	
Cinnamic acid derivative																				+				
Methoxycinnamic acid															+									
Cinnamic acid-*O* xylosyl hexoside																				+				
Hydrocinnamic acid	+																							
Hydrocinnamic acid glucoside																					+			
Hydroxycinnamoyl-quinic acid																						+		
*p*-Coumaric acid		+					+					+			+	+	+			+	+		+	+
*p*-Coumaricacid derivate												+												+
Coumaroyl quinic acid		+	+		+							+												+
Coumaroyl quinic acid derivative		+																						
Coumaric acid hexoside																				+	+			
Chicoric acid																					+			
Ferulic acid	+	+					+								+	+	+	+	+	+	+		+	+
Ferulic acid derivative																					+		+	
3-*O*-Feruloylquinic acid		+	+																	+				+
Feruloylquinic glucoside		+	+		+																			
Feruloyl-glucoside																	+							
Hydroxy-ferulic acid hexoside			+																					
Hydroxy-ferulic acid rhamnoside					+																			
Feruloyl tartaric acid (fertaric acid)		+	+		+																+			
6′-*O*-Sinapoylsucrose			+		+																			
3,4-Dihydroxyphenyllactic acid hexoside																					+			
3-(3,4-Dihydroxyphenyl)-2-hydroxypropanoic acid																					+			
Rosmarinic acid			+														+				+	+	+	
Rosmarinic acid hexoside																					+		+	
Rosmarinic acid-3-*O*-glucoside																	+						+	
Dihydroxy-dihydro feruloyl methyl rosmarinic acid		+	+		+																			
Methylrosmaric acid		+	+		+												+				+		+	
*p*-Hydroxybenzylrosmarinic acid			+		+																			
Isosalvianolic acid A																					+			
Methyl melitrate																					+			
Salvianolic acid A																								
Salvianolic acid B																					+			
Salvianolic acid C																					+			
Sinapaldehyde																					+			+
3-Sinapoylquinic acid			+																					
Sinapic acid																					+		+	+
Yunnaneic acid F																					+			
Verbascoside																							+	
Class: Phenolic acid. Sub Class: Hydroxyphenylacetic acids																								
p-Hydroxyphenylacetic acid															+					+				
3,4-Dihydroxyphenylacetic acid																					+		+	
3,4-dihydroxyphenyllactic acid hexoside																					+			
Class: Flavonoid. Sub Class: Flavanones																								
Eriodictyol																					+		+	
Eriodictyol 7-*O*-rutinoside																	+						+	
Eriodictyol-7-*O*-glucuronide																							+	
Eriodictyol-*O*-di-hexoside																						+	+	
Eriodictyol-*O*-hexoside																							+	+
Eriodictyol-*O*-hexuronide																						+		
Glucodistylin																+								
Glucodistylin isomer																+								
Hesperetin 7-*O*-rutinoside (Hesperidin)								+									+				+		+	+
Methyleriodictyol-*O*-pentosylhexoside																						+		
Naringenin					+										+	+				+	+	+	+	+
Naringenin-di-hexoside																						+		
Naringenin 7-*O*-glucoside (Naringin-Prunina)		+	+		+			+									+			+		+	+	+
Pinocembrin																								+
Class: Flavonoid. Sub Class: Flavones																								
Apigenin	+	+	+				+	+	+								+				+	+	+	+
Apigenin 7-*O*-glucuronide																					+		+	
Apigenin glucuronide hexoside																					+	+	+	
Apigenin 6,8-di-*C*-glucoside	+		+		+																		+	
Apigenin 7-*O*-glucoside							+									+	+				+	+	+	
Apigenin *C*-hexoside							+														+			
Apigenin pentoside		+																		+				
Apigenin-7-*O*-rutinoside																	+							
Apigenin-*O*-hexoside							+															+		
Apigenin-*O*-hexoside derivative																				+				
Isovitexin 7-*O*-glucoside																				+				
Apigenin 8-*C*-glucoside (Vitexin)																	+			+				
2″-*O*-pentosyl-6-*C*-hexosyl-apigenin							+																	
2″-*O*-Pentosyl-8-*C*-hexoside apigenin isomer I							+																	
2″-*O*-Pentosyl-8-*C*-hexoside apigenin isomer II							+																	
2″-*O*-pentosyl-8-*C*-hexosyl-apigenin							+																	
2″-*O*-Pentoxide-8-*C*-hexoside apigenin							+																	
4′-*O*-Rutinoside of 7-*O*-methylated apigenin																							+	
6″-*O*-(3-hydroxy-3-methylglutaroyl)-2″-*O*-pentosyl-C-hexosyl-apigenin							+																	
Apigenin 4′-methyl ether	+		+														+							
Apigenin 7-methyl ether (Genkwanin)	+		+																		+			
Apigenin 4′,7-dimethyl ether	+		+														+				+			
Chalcone															+									
Chrysin derivative							+																	
Chrysin-7-*O*-glucoside							+																	
Circimaritin																								
Hispidulin (Scutellarein 6-methyl ether)																	+						+	
Hispidulin 7-*O*-glucose (homoplantaginin)																	+							
6″-*O*-(*E*)-feruloylhomoplantaginin																	+							
Hispidulin-rutinoside																	+							
Hypolaetin di-glucuronide																					+			
Hypolaetin 8-*O*-glucuronide																					+			
Isoscutellarein 7-*O*-glucoside																	+							
Isoscutellarein 8-*O*-glucuronide																					+			
Ladanein (5,6-dihydroxy-7,4′-dimethoxyflavone)																	+							
Luteolin (3′.4′.5.7-Tetrahydroxyflavone)		+							+						+	+	+				+	+	+	+
Chrysoeriol-*O*-hexoside (Luteolin 3′-methyl ether)																						+		
Diosmetin (Luteolin 4′-methyl ether)																	+							
Cirsilineol (6-Methoxyluteolin 3′,7-dimethyl ether)																							+	
Luteolin 7,3-dimethyl ether																					+			
Luteolin 3′-*O*-glucuronide																	+							
Luteolin-7-*O*-glucuronide																	+				+		+	+
Luteolin 7,4′-di-glucuronide																					+		+	
Luteolin 4-*O*-glucoside																	+							
Luteolin 7-*O*-glucoside							+								+		+				+		+	
Luteolin-5-*O*-glucoside							+																	
Luteolin 8-*C*-glucoside (Orientin)							+	+																
Luteolin 6-*C*-glucoside (Isoorientin)																	+							
Luteolin-*O*-hexorunide																						+		
Luteolin-7-*O*-rutinoside		+	+		+										+		+						+	
Luteolin-hexoside																	+					+		
Luteolin 6-hydroxy-7-*O*-glucoside																	+							
Luteolin-*O*-malonyl-hexoside)		+	+				+																	
2″-*O*-Pentosyl-8-*C*-hexoside luteolin							+																	
2″-*O*-pentosyl-6-*C*-hexosyl-luteolin							+																	
2″-*O*-pentosyl-8-*C*-hexosyl-luteolin							+																	
6″-*O*-(3-hydroxy-3-methylglutaroyl)-2″-*O*-pentosyl-*C*-hexosyl-luteolin							+																	
Nepitrin (Nepetin 7-*O*-glucoside)																	+							
6″-*O*-(*E*)-feruloylnepitrin																	+							
Salvigenin (5-Hydroxy-6,7,4′-trimethoxyflavone)																								
Techtochrysin																	+							
Class: Flavonoid. Sub Class: Flavonols																								
Isorhamnetin					+											+								
Isorhamnetin 3-*O*-glucoside																	+	+		+	+			
Isorhamnetin 3-*O*-rutinoside		+															+				+			
Isorhamnetin-3-*O*-hexoside							+				+						+							
Isorhamnetin-*O*-(6″-caffeoyl)hexoside											+													
Isorhamnetin-*O*-deoxyhexosyl-hexoside							+																	
Isorhamnetin-*O*-hexoside-*O*-rhamnoside	+										+													
Galangin (3,5,7-Trihydroxyflavone)																								+
Kaempferol					+			+		+	+	+			+	+	+			+	+	+	+	+
6-Hydroxykaempferol																				+				
Dihydrokaempferol 3-*O*-glucoside																							+	
Kaempferol 3-methyl ether	+		+		+																			
Kaempferol 3 4′-dimethyl ether	+		+																					
Kaempferol 3 7-dimethyl ether	+		+																					
kaempferol 3,7,4′-trimmethyl ether	+		+																					
kaempferol methylether *O*-rutinoside																							+	
Kaempferol dimethylether hexoside	+																							
Kaempferol 3-*O*-(6″-galloyl) glucoside											+							+		+				
kaempferol-3-*O*-(6″-feruloyl)-β-D-glucopyranoside																								+
Kaempferol 3-*O*-(6″-p-coumaroyl) glucoside					+															+				
kaempferol-3-*O*-(2′′,6′′-di-p-coumaroyl)glucoside isomers																		+						
kaempferol-3-*O*-(2″,6″-di-E-p-coumaroyl)-glucopyranoside																		+						
kaempferol-3-*O*-(3′′-acetyl-2″,6′′-di-p-coumaroyl)glucoside																		+						
kaempferol-3-*O*-(3′′,4′′-diacetyl-2′′,6′′-di-p-coumaroyl)glucoside isomers																		+						
kaempferol-3-*O*-(6″-p-coumaroyl) glucopyranoside(Tiliroside)																		+		+				+
Kaempferol-3-galactoside-6″-rhamnoside-3′″- rhamnoside					+																			
kaempferol malonyl glucoside																								+
kaempferol 3-*O*-ramnopyranoside																				+				
Kaempferol-*O*-rhamnoside		+			+								+							+				
Kaempferol 7-*O*-(6″-rhamnosyl) glucoside		+								+										+				
kaempferol 3-*O*-arabinofuranoside																				+				
kaempferol-3-*O*-arabinopyranoside																				+				
Kaempferol 3-*O*-glucoside (Astragalin)	+	+																		+				+
Kaempferol 3-*O*-rutinoside	+	+													+									+
Kaempferol-acetyl-*O*-rutinoside		+																						
kaempferol-acetyl-*O*-rahmnoside												+												
Kaempferol-acetyl-*O*-hexoside		+					+	+																+
Kaempferol *O*-glucuronide																							+	+
Kaempferol 7-*O*-hexuronide																				+				
Kaempferol 3-*O*-pentoside		+																		+				
Kaempferol *O*-hexoside							+	+		+										+				+
Kaempferol *O*-pentosyl hexoside		+																						+
Kaempferol-3-*O*-hydroxybenzoyl glucoside																								+
kaempferol-3-*O*-galactoside																								+
Kaempferol xyloside																				+				
Kaempferol-*O*-di-hexoside	+																							+
Morin																				+				
Myricetin (3.3′.4′.5.5′.7-Hexahydroxyflavone)	+	+									+	+			+	+		+	+	+				
Myricetin 3-*O*-(6″-rhamnosyl) glucoside																				+				
Myricetin-*O*-(6″-benzoyl) hexoside											+													
Myricetin-*O*-(6″-cinnamoyl) hexoside											+													
Myricetin-*O*-(6″-p-coumaroyl) hexoside											+													
Myricetin-*O*-(galloyl) hexoside		+	+		+															+				
Methoxy-myricetin-*O*-rhamnoside											+													
Myricetin 3,7,4′,5′-tetramethyl ether			+																					
Myricetin-3-arabinoside														+										
myricetin 3-*O*-arabinofuranoside																				+				
Myricetin-3-*O*-galactoside														+										
Myricetin-3-*O*-glucoside												+			+	+				+				
Myricetin-3-*O*-glucuronide		+													+	+								
Myricetin 3-*O*-hexoside		+			+					+	+		+							+				
Myricetin 7-*O*-hexuronide																				+				
Myricetin 3-*O*-pentoside																				+				
Myricetin 7-*O*-pentoside											+		+							+				
Myricetin-*O*-rhamnoside (Myricitrin)	+	+	+		+						+	+	+	+	+	+		+		+				
Myricitrin-2′′-*O*-gallate (Desmanthin)																+								
Myricetin-*O*-rutinoside		+			+										+	+		+						
Myricetin 3-*O*-xyloside															+					+				
Pinobanksin (bioflavonoide)																								+
Quercetin (3.3′.4′.5.7-Pentahydroxyflavone)	+	+	+		+		+	+		+					+	+	+	+			+	+	+	+
Quercetin-(acetyl) rutinoside		+																						
Quercetin-(acetyl) hexoside		+					+																	
Quercetin-(acetyl)-*O*-rhamnoside												+												
Quercetin-*O*-(6″-cinnamoyl) hexoside										+	+													
quercetin 3-*O*-(2′-coumaroyl) rutinoside		+		+	+																			
Quercetin 3-*O*-(6″-p-coumaroyl) hexoside																				+				
Quercetin 3-*O*-(6″-galloyl) hexoside																				+				
Quercetin-*O*-(6″-p-hydroxybenzoyl) hexoside											+									+				
Quercetin-*O*-(malonyl) hexoside											+													
Quercetrin-*O*-gallate															+	+								
Quercertin methyl ether-3-*O*-galactoside		+																						
Quercetin 4′,5′-dimethyl ether										+														
Quercetin 3,7,4′,5′-tetramethyl ether			+																					
Quercetin 3-*O*-arabinoside																				+				
quercetin 3-*O*-arabinofuranoside															+					+				
Quercetin 3-*O*-galactoside (Hyperoside)															+					+			+	+
Quercetin 3-*O*-glucoside (Isoquercetin)	+	+	+	+		+	+	+				+		+	+			+		+	+	+	+	+
Quercetin 3-*O*-glucuronide																+				+			+	+
Quercetin-*O*-hexoside							+			+	+	+	+							+		+		+
Quercetin 3-*O*-hexuronide																				+		+		
Quercetin hexose protocatechuic acid																				+				
Quercetin-*O*-rhamnoside (Quercetrin)	+	+	+		+							+	+	+	+	+				+				+
Quercetin 3-*O*-rutinoside (Rutin)	+	+	+		+		+	+		+		+			+	+	+	+		+	+		+	+
Quercetin 3-*O*-pentoside	+	+											+							+				
Quercetin 3-*O*-xyloside																				+				
Quercetin 3-*O*-rhamnoside-7-*O*-glucoside																				+				
Quercetin 3,4-diglucoside	+		+		+										+									
Quercetin-*O*-deoxyhexosyl-hexoside							+																	
Quercetin-*O*-dihexoside	+						+																	
Quercetin-pentosyl-hexoside	+	+																						
Taxifolin (dihydroquercetin)															+	+							+	
Taxifolin-3-*O*-glucoside																+								+
Taxifolin 3-*O*-rhamnoside																				+				
Taxifolin-*O*-hexoside											+													+
Taxifolin pentoside																								+
Class: Flavonoid. Sub Class: Flavanols																								
Catechin	+	+										+			+	+		+	+	+	+		+	+
Catechin 3-gallate																				+				
4,3′,4′-Trimethylcatechin			+		+																			
Catechin derivates												+												
Catechin glucose																				+				
Catechin-( 4α→8)-Catechin (Procyanidin B3)																		+		+				
Dehydrodicatechin A																		+						
Epicatechin		+			+							+			+			+	+	+	+		+	+
Epicatechin derivatives																				+				
Epicatechin methyl gallate		+																						
Epicatechin gallate		+													+			+		+	+			
epicatechin-4,6-catechin																				+				
epicatechin-4,8-catechin																				+				
epicatechin-4,8-epicatechin-4,8-catechin																				+				
Epicatechin-4,8-epicatechin-4,8-Epicatechin																				+				
Epicatechin-A-epicatechin		+														+				+				+
Epicatechin-B-epicatechin-A-epicatechin																								+
Epicatechin-epicatechin 3-*O* gallate																				+				
Epicatechin-epigallocatechin				+	+															+				
Epigallocatechin	+			+	+															+				
Epigallocatechin gallate(Teatannin II)		+														+				+				+
Epigallocatechin–catechin																		+						
Epigallocatechin–epicatechin																				+				
Epigallocatechin–epigallocatechin	+				+															+				
Fzelechin-catechin-3-*O*-rhamnoside (proanthocyanidin)																		+						
Gallocatechin																+	+			+				
Gallocatechin-4,8-catechin																				+				
Procyanidin B2		+																						
Tannic acid															+									
Class: Flavonoid. Sub Class: Anthocyanins																								
Cyanidin 3-*O*-arabinoside															+					+				
Cyanidin-3-galactoside																				+				
Cyanidin 3-*O*-glucoside		+	+											+	+					+				+
Cyanidin-3-*O*-rutinoside																				+				
Cyanidin 3-*O*-xyloside																								+
Cyanidin dihexoside																								+
Cyanidin-3,5-diglucoside																				+				
Delphinidin 3-*O*-galactoside																				+				
Delphinidin 3-*O*-glucoside														+	+					+				+
Malvidin-3-*O*-glucoside/Oenin		+			+									+						+				
Pelargonidin 3-*O*-(6″-malonyl) glucoside		+			+																			
Pelargonidin-3-*O*-glucoside																								+
Pelargonidin 3-rutinoside																								+
Peonidin 3-*O*-(6″-*p*-coumaroyl) glucoside		+			+																			
Peonidin-3-*O*-glucoside														+										
Petunidin		+			+																			
Petunidin-3-*O*-glucoside		+												+										
Class: Flavonoid. Sub Class: Isoflavonoids																								
Daidzein																								
3′-Hydroxydaidzein									+															
Genistein									+						+									
2′-Hydroxygenistein									+															
Glycitin 6″-*O*-malonate			+		+																			
Class: Other polyphenols. Sub Class: Hydroxybenzaldehydes																								
4-hydroxybenzaldehyde																	+				+			
4-hydroxybenzoic acid 4-(6-*O*-sulfo)glucoside																					+			
Syringaldehyde								+																+
Vanillin	+							+															+	
Class: Other polyphenol. Sub Class: Hydroxycoumarins																								
4-methylumbelliferone																								+
6,7-Dihydroxycoumarin 3*O*-glucoside (Aesculin)																				+				
Coumarin																	+							
Class: Other polyphenol. Sub Class: Tyroslos																								
Oleuropein		+																						
Class: Lignans. Sub Class: Lignans																								
Carnosic acid																	+						+	
Carnosol																	+					+	+	
Isolariciresinol 3-glucoside																					+			
Methyl carnosic acid																	+							
Pinoresinol																					+		+	
Rosmanol																	+							
Rosmanol derivate																	+							
Sagerinic acid																					+			
Thymol																							+	
Class: Other polyphenols. Sub Class: Other polyphenols																								
5-Nonadecylresorcinol																					+			
Arbutin																				+				
Catechol																	+	+	+		+			
Coniferaldehyde																				+				+
Hydroquinone derivative																				+				
Salvianolic acid																							+	
Salvianolic acid A																					+	+		
salvianolic acid B (lithospermic acid B)																					+	+	+	
Salvianolic acid B/E/L																							+	
Salvianolic acid C																					+			
Salvianolic acid C isomer																					+			
Salvianolic acid F																						+		
Salvianolic acid K																						+		
Salvianolic acid I																						+		
Sculetin																					+			
Class: Stilbenes. Sub Class: Stilbenes																								
Piceid																				+				
Resveratrol																+				+				+
Stilbericoside																				+				
References	[75,78,79,80,81,82,83,84,85,86]	[81,87,88]	[81,83,88,89,90]	[81]	[81,88,91]	[81,82]	[92,93,94,95,96]	[97,98,99]	[100]	[100]	[101]	[102,103]	[101]	[83,104]	[105,106,107]	[108,109]	[110,111,112,113,114,115]	[116,117,118]	[117]	[119,120,121,122,123,124,125,126,127,128,129,130,131,132,133]	[134,135,136,137,138,139]	[140,141,142]	[143,144,145,146,147,148,149,150,151,152]	[92,153,154,155,156,157,158,159,160,161,162]

Cl: *C. ladanifer*; Cs: *C. salviifolius*; Cm: *C. monspeliensis*; Cc: *C. crispus*; Ca: *C. albidus*; Cp: *C. populifolius*; Cym: *C. multiflorus*; Cys: *C. scoparius*; Cyst: *C. striatus*; Em: *E. multiflora*; Es: *E. scoparia*; Ea: *E. australis*; Cv: *C. vulgaris*; Mc: *M. communis*; Pl: *P. lentiscus*; Pt: *P. terebinthus*; Ro: *R. officinalis*; Qi: *Q. ilex*; Qs: *Q. suber*; Au: *A. unedo*; Ls: *L. stoechas*; Tm: *T. mastichina*; Tv: *T. vulgaris*; Ru: *R. ulmifolius*.

There is a clear difference in the number of compounds identified in each species. The species with the largest number of compounds is *A. unedo* (142 compounds), whereas only 5 to 8 compounds have been identified in *C. crispus*, *C. striatus* and *C. vulgaris*. The identification of more or fewer compounds in a species is due to the number of studies conducted on each, which is determined by their commercial interest. Some of them, such as *A. unedo*, have a high commercial interest, which explains the existence of many studies on this species and, therefore, a larger number of compounds identified in it.

Furthermore, the compounds are unequally represented. Some phenols have only been cited in one species, while others have been reported in many species (Table 1). Considering the compounds that appear in more than 5 species (Table 3), 38 phenolic compounds are found in these species, 15 of which are phenolic acids and 23 are flavonoids.

## 4. Neuroprotective Effect of the Most Represented Phenolic Compounds in the Selected Species

The most distributed phenolic compounds among the selected species belong to two classes (Table 3): phenolic acids and flavonoids. Activity against neurodegenerative disorders has been attributed to most of these compounds. In fact, one of the activities most strongly associated with phenolic acids is their antioxidant capacity. This activity depends on the number of hydroxyl moieties attached to the aromatic ring of benzoic or cinnamic acid molecules. For example, Rice-Evans et al. [163] reported that the total antioxidant activity of phenolic acids, in decreasing order, is gallic > *p*-coumaric > ferulic > vanillic > syringic > caffeic > *m*-coumaric > protocatechuic > gentisic > o-coumaric > salicylic > *p*-hydroxybenzoic. Free-radical scavenging is the activity that confers them with the protective function against neurodegenerative disorders.

Six phenolic acids (gallic, chlorogenic, ferulic, caffeic, vanillic and *p*-coumaric acids) are represented in more than 40% of the species studied. These compounds have been assigned neuroprotective functions (Figure 2). Some of the properties attributed to them are described below.

Gallic acid (GA) is present in 70% of the species that dominate the ecosystems of Extremadura. GA is a well-known 5,4,3-trihydroxybenzoic acid found abundantly in free and conjugated (hydrolyzable tannins) or esterified forms in many plants [164]. It is a phenol with great interest for the treatment of patients with AD and PD, due to its antioxidant, anti-inflammatory, and anti-amyloidogenic properties [165]. Different studies have shown its application as a therapy to interact with amyloid (Aβ) monomers and fibrils. These studies have proved that GA-loaded transferrin-functionalized liposomes could inhibit Aβ_1–42_ aggregation and fibrillation and disrupt preformed fibrils, and thus it could be considered for AD therapy [166]. GA has been demonstrated to reduce memory deficit and cerebral oxidative stress in a unilateral 6-hydroxydopamine-induced PD model in rats [167]. Moreover, its neuroprotective effect has been shown in models of traumatic brain injury [168] and glutamate-induced neurotoxicity in rats [169], due to the improvement in the antioxidant profile and the inhibition of proinflammatory cytokine generation [168,169].

The mechanisms by which GA exerts its prophylactic action in these processes have been analyzed in several studies. For instance, refs. [170,171] reported that, in animals with multiple sclerosis (MS), the administration of GA improved the oxidative and inflammatory response and induced dendritic hyperplasia. This causes an increase in the number of dendritic spines, which could explain the positive response in the dendritic morphology of the three regions (CA1-CA3-DG) of the rat hippocampus with MS. It has been indicated that GA inhibits Aβ-induced neurotoxicity via suppressing microglial-mediated neuroinflammation and decreasing cytokine generation [172]. Studies conducted by [173] show that GA treatment maintains Ca^2+^ homeostasis and insulin-like growth factor 1 (IGF-1) expression and protects neurons from glutamate-induced neurotoxicity.

A recent study conducted by [174] estimated the neuroprotective effects of (GA) against aluminum chloride-induced AD in adult Wistar rats. The trials performed showed that there was a significant decrease in antioxidant enzymes, serum electrolyte and neurotransmitter levels with a corresponding increase in stress markers (MDA, H_2_O_2_ and NO) among the rats treated with aluminum, which were restored to nearly normal levels after GA administration. Histological observation showed neurofibrillary tangles and amyloid plaques in the external granular layer of the rats treated with aluminum, although this effect disappeared after GA administration [174].

These studies suggest that structural and functional alterations in the neurons of animals with neurodegenerative diseases are reverted after GA treatment; consequently, neurochemical processes are restored, improving recognition memory [175].

Chlorogenic acid (CGA) is another type of polyphenol that has demonstrated potent anti-inflammatory and antioxidant activities [176]. CGA is present in 54.1% of the species analyzed in this study. Its mechanism of action could be related to the attenuation of mRNA and protein expression levels of proinflammatory and profibrotic mediators, and the reduction of the levels of serum proinflammatory cytokines, such as TNF-α (tumor necrosis factor-alfa) IL-6 (interleukin-6) and IL-1β, as is reported in studies conducted in female rats [177]. CGA treatment also suppressed CCl_4_-induced NF-κB (nuclear factor kappa-B) activation and reduced the expression levels of Toll-like receptor 4, myeloid differentiation factor 88, inducible nitric oxide synthase and cyclooxygenase in rats exposed to CCl_4_ [178].

Ferulic acid (FA) is present in 50% of the 24 species selected in this study. This compound has been reported to increase neuronal survival, enhance antioxidant enzyme function, modulate multiple neuronal signal transduction, and impair cholinesterase activity (ChAT) [179].

FA has been identified as an effective ROS and RNS scavenger, reducing the likelihood of attack of radicals on proteins and thereby preventing oxidative changes. The antioxidant and anti-inflammatory potential of FA could be due to its ability to suppress leukotriene synthesis and reduce oxidative stress in the brain [180].

Several studies have highlighted the anti-inflammatory effects of FA [181,182]. Particularly, FA has been shown to reduce the neuroinflammation induced by chronic unpredictable mild stress in the prefrontal cortex through the inhibition of NF-κB activation [183].

The potential role of FA against AD has also been investigated in cell models. In particular, Kikugawa et al. [184] showed that the pretreatment of primary cerebral cortical neurons with FA exerted a protective effect toward Aβ_25–35_-induced cytotoxicity; moreover, FA was able to inhibit the aggregation of Aβ_25–35_, Aβ_1–40_ and Aβ_1–42_ and to destabilize pre-aggregated Aβ.

It is worth highlighting that the potential usefulness of FA in AD has also been investigated in in vivo studies [185]. Yan et al. [186] reported that IL-1β production, neuroinflammation and gliosis, induced by the intracerebroventricular injection of Aβ in the mouse hippocampus, were counteracted by pretreatment with FA, and this phenolic acid was able to improve memory loss. Kim et al. [41] also showed that FA prevented the Aβ_1–42_-induced increase in endothelial nitric oxide synthase and 3-nitrotyrosine and suppressed IL-1α immunoreactivity in the hippocampus [187]. Along with the amelioration of Aβ plaque deposition, Wang et al. [188] recently found that FA prevented the reduction in the density and diameter of hippocampal capillaries, thus favoring the oxygen and nutrient supply and the removal of metabolic wastes from the brain, which finally led to improved spatial memory.

Caffeic acid (CA) is another phenol that is present among 50% of the selected species. It has been shown that CA reduces elevated oxidative stress and neuroinflammation and improves synaptic/memory dysfunctions in AD mice [189]. Studies conducted in mice have reported that CA has strong antioxidative and anti-inflammatory properties and prevents the mice brain from A*B*-induced oxidative stress and neuroinflammation [190]. These findings suggest that CA significantly reduces activated microglia and astrocytes in the brains of AD mice.

There are markers clearly related to neurodegenerative conditions and memory dysfunctions, such as phosphatidylinositol 3-kinase /protein kinase b signaling pathway, and downregulation of neuronal growth factors, such as brain-derived neurotrophic factor [191,192,193]. It has been proved that CA considerably upregulates the expression of these markers in the brains of Aβ-injected mice, and a significant improvement was observed with CA treatment [194].

Vanillic acid (VA) is present in 42% of the species analyzed in this study, and this phenolic acid has been reported to have a clear anti-inflammatory function [195]. Studies conducted with VA have shown that this compound significantly increases neurite outgrowth after 48 h in culture. This compound significantly reduces the expression of cyclooxygenase-2, NF-κB, tenascin-C, chondroitin sulfate proteoglycans and glial fibrillary acidic protein in astrocytes in the LPC-induced model of inflammation. This study supports the hypothesis that VA has anti-inflammatory activities, and, since axonal and synaptic damage is present in most and possibly all neurodegenerative diseases, including AD, PD, and HD [196], the effects of VA on neurite outgrowth make it a potential candidate to encourage the regeneration of neurites after demyelination.

*p*-coumaric acid (PCA) is present in over 40% of the studied species. In recent years, this compound has been the focus of numerous studies due to its wide variety of biological activities: antioxidant [197], anti-inflammatory [198], neuroprotective [199] and memory-ameliorating effects [200]. Authors such as Rashno et al. [201] explored the effects of oral administration of PCA on passive avoidance memory function, LTP (long-term potentiation) induction in the hippocampal dentate gyrus and hippocampal Aβ plaque formation following AlCl_3_ exposure in male rats, a condition that resembles the symptoms of AD. The results obtained by this group demonstrated that treatment with PCA alleviated passive avoidance deficit, improved hippocampal LTP impairment and prevented Aβ plaque formation in the AlCl_3_-exposed rats. Cognitive-improving effects of PCA have been reported in various neuropathological conditions, such as cerebral ischemia [202], lipopolysaccharide-induced neurological changes [200] and scopolamine-induced neurotoxicity [42].

In addition to the group of phenolic acids, the other group of phenols widely distributed among these species is that of flavonoids. Of the 224 different flavonoids that can be found in the entire set of species selected in this study, 23 are present in over 20% of them. Within this group, quercetin and its derivatives quercetin 3-O-rutinoside and quercetin 3-O-glucoside (isoquercetin) stand out, as they are present in over 60% of the selected species. Other flavonols and flavones that are also widely distributed include quercetin-O-rhamnoside (quercetrin), apigenin, kaempferol and myricetin-O-rhamnoside (myricitrin), being present in 45–55% of the selected species.

Different in vitro and in vivo experiments found that these polyphenols may exert a beneficial effect in the prevention and treatment of neurodegenerative diseases associated with oxidative stress, shown in Figure 3, and that the activity of flavonoids such as galangin, kaempferol, quercetin, myricetin, fisetin, apigenin, luteolin and rutin was correlated with the number of OH groups and their side on their phenyl ring [63,203]. It is worth highlighting that these phenolic compounds and their metabolites can enter the brain at detectable levels in mammals, which supports their direct neurological action [204].

Flavonoids, depending on the degree of oxidation and saturation in the heterocyclic C-ring, can be divided into different subclasses, varying in their bioavailability. Thus, flavanols, flavanones and flavonol glycosides have intermediate rates of absorption and bioavailability, while proanthocyanidins, flavanol gallates and anthocyanins have the lowest absorption [205]. According to different studies, epicatechin metabolites seem to reach the brain of rodents at levels that might be physiologically effective [206], and some conjugated forms of quercetin can also accumulate in the brain after oral administration [207,208].

Among flavonoids, flavonols and flavones constitute the largest group and have been associated with a clear neuroprotective role [70,209,210,211,212,213]. It has been demonstrated in numerous studies that flavonols such as quercetin, myricetin and kaempferols, as well as their derivatives, have strong antioxidant activity [209] and also demonstrate that their glucosylated derivatives have greater activity than the corresponding aglycones [213]. The radical scavenging and metal chelating activity of flavonols contribute to ameliorating oxidative stress [167,214]; in turn, this activity depends on the number of the hydroxyl and the sugar moiety associated [213,215].

In addition to the antioxidant capacity of these compounds as free-radical scavengers, the mechanisms involved in the neuroprotector effect of these compounds would be associated with their capacity to inhibit Aβ aggregation, the amyloid precursor protein cleaving enzyme (BACE1) [216] and AChE [217]. Studies on AChE inhibition in the brain of oxidative stress-induced rats report that AChE activity significantly decreases [218,219]. Specifically, treatment with flavonol quercetin in hippocampal neurons has resulted in the elevation of neurogenesis, synaptogenesis, and cell proliferation, as well as restoration of Aβ-induced synaptic loss [220]. This flavonol also exerts positive effects on PD, as it can inhibit the activity of catechol-O-methyltransferase and monoamine oxidase enzymes, which can lead to an increase in the bioavailability of L-dopa in the brain [4]. Quercetin has also been attributed to the capacity to act through different signaling pathways, including regulation of cytokines via Nrf2 (nuclear factor erythroid-derived 2), JNK (Jun-NH_2_-terminal kinase), protein kinase C, MAPK (the mitogen-activated protein kinase) signaling cascades, and PI3K/Akt (phosphoinositide 3-kinase) pathways [221].

Another flavonoid with clear antioxidant functions is flavan-3-ol catechin [222], which, along with its isomers and/or conjugates of gallic acid, is a naturally occurring constituent in plants [223]. This has been observed among the studied species, as this flavonoid is present in 46% of the species. Different studies report the neuroprotective properties of catechins, mostly through antioxidative and anti-inflammatory effects, mainly involving Nrf2 and NF-kB signaling pathways [222,224,225]. One in vivo study has revealed that it can improve cognitive impairment induced by doxorubicin via increasing antioxidant defense, preventing neuroinflammation and inhibiting AChE [226]. Catechin has also been indicated to inhibit the late stages of Aβ-soluble aggregate growth change in the fibrillar form of Aβ [227]. It has also prevented neurotoxin-induced dopamine neuron loss in substantia nigra in a mouse model of PD [228]. The other flavanol with a high representation among the studied species is epicatechin, which is present in 42% of them. It has been demonstrated that epicatechin treatment prevents oxidative damage to the hippocampus induced by Aβ_25–35_ [229]. This flavanol may reduce Tau hyperphosphorylation, downregulate BACE1 and Aβ_1–42_ expression and boost AD rats’ antioxidant system, as well as their cognition and memory [230,231].

## 5. Main Species with Neuroprotective Activity

As can be observed, the species considered in this study can be an important source of phenolic compounds with activity against neurodegenerative diseases. However, focusing on the most represented compounds (in over 40% of the analyzed species) and the species that contain all or most of these compounds (Table 4), 7 of these species stand out: *C. multiflorus*, *P. lentiscus*, *A. unedo*, *L. stoechas*, *R. ulmifolius* and *T. vulgaris* contain the 6 most frequent phenolic acids (gallic, chlorogenic, ferulic, caffeic, vanillic and *p*-coumaric acids) and *C. salviifolius* and *P. lentiscus* contain the main flavonoids.

Of these 7 species, studies have been conducted with extracts of *P. lentiscus*, *L. stoechas* and *T. vulgaris* to demonstrate their activity against neurodegenerative diseases [107,136,138,151,232]. These studies have reported the in vitro AChE inhibitory activity of *P. lentiscus* and its extract exhibited a significant dose-related AChE inhibitory activity. This extract also showed the ability to prevent neurodegeneration and improve memory and cognitive function. This indicates that *P. lentiscus* inhibited Al-induced neurodegeneration of neurons in the brain cortex, which is known to be susceptible in AD and to play an important role in learning and memory functions [107,233,234]. These findings explain the protective effects of *P. lentiscus* on cognitive deficit. Moreover, the capability of the extract to correct the in vivo disorders may be explained by its ability to inhibit oxidative stress and lipid oxidation induced by Al [213,235].

The extracts of *L. stoechas* also significantly (*p* < 0.001) enhanced the retention power and learning capacity of mice brains. Similarly, treatment of animals with extracts of la- vender showed a significant (*p* < 0.001) reduction in the level of AChE and relieved the patient of memory loss [136,139].

On their part, studies conducted with the extract of *T. vulgaris* also indicate that this species can present neuroprotective effects [151,152]. The results obtained by [236] suggest that the antiamnesic effect of *T. vulgaris* extract on scopolamine-induced memory impairment may be related to the antioxidant activity of the extract or mediation of the cholinergic nervous system [148,150].

The protective activities attributed to these species can be inherent to the presence of these phenolic compounds (flavonoids and phenolic acids), where the presence of all of them may exert a synergistic effect as neuroprotective agents.

## 6. Conclusions

This review highlights the relevance of the species of Mediterranean ecosystems as a diverse source of phenolic compounds. Among these compounds, phenolic acids and flavonoids stand out. The most represented compounds among the species studied are gallic, vanillic, caffeic, chlorogenic, *p*-coumaric and ferulic acids, apigenin, kaempferol, myricitrin, quercetin, isoquercetin, quercetrin, rutin, catechin and epicatechin, which have been attributed neuroprotective functions. Given this information, these Mediterranean scrub species could be considered as sources of compounds for use in therapy against neurodegenerative diseases such as AD and PD.

## Figures and Tables

**Figure 1 molecules-28-08133-f001:**
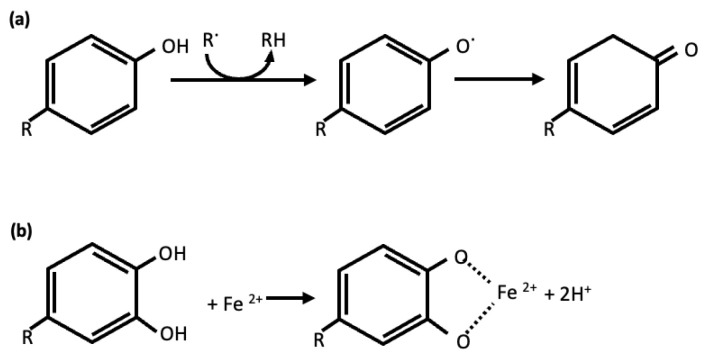
Mechanism of scavenging of ROS (**a**) and metal chelation (**b**) by phenolic compounds antioxidants.

**Figure 2 molecules-28-08133-f002:**
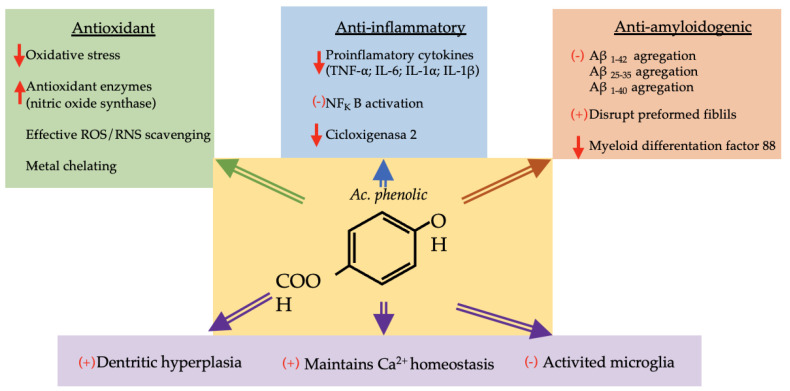
Diagram with neuroprotective mechanisms of phenolic acid. ↑: increase, ↓: decrease, (+): activation, (−): inhibition, ROS: reactive oxygen species, RNS: reactive nitrogen species, TNF-α: tumor necrosis factor-alfa, IL: interleukin, NF_k_B: nuclear factor kappa B, AB: amyloid beta-peptide.

**Figure 3 molecules-28-08133-f003:**
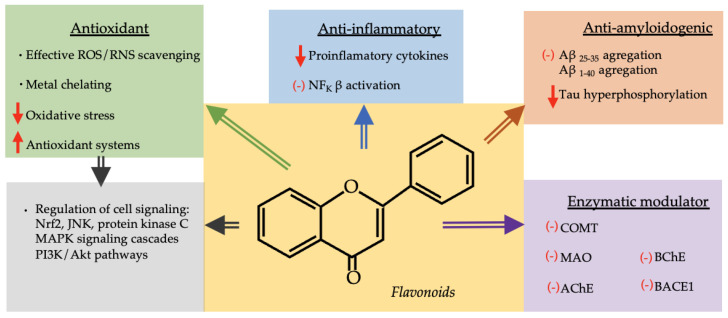
Diagram with neuroprotective mechanisms of flavonoids. ↑: increase, ↓: decrease, (−): inhibition, ROS: reactive oxygen species, RNS: reactive nitrogen species, NF_k_B: nuclear factor kappa B, AB: amyloid beta-peptide, Nrf2: nuclear factor erythroid-derived 2, JNK (Jun-NH_2_-terminal kinase), MAPK (the mitogen-activated protein kinase), PI3K/Akt (phosphoinositide 3-kinase), COMT (catechol-O-methyltransferase), MAO (monoamine oxidase), AChE (acetylcholinesterase), BchE (butyrylcholinesterase), BACE1 (amyloid precursor protein cleaving enzyme I).

**Table 2 molecules-28-08133-t002:** Number and percentage of phenolic compounds, grouped by class and subclass, found in the 24 species selected in this study.

	N° Compound	Percentage (%)
Class: Phenolic acid	144	35.72%
Sub Class: Hydroxybenzoic acids	82	20.34%
Sub Class: Hydroxycinnamic acids	59	14.64%
Sub Class: Hydroxyphenylacetic acids	3	0.74%
Class: Flavonoid	224	55.56%
Sub Class: Flavanones	13	3.22%
Sub Class: Flavones	59	14.64%
Sub Class: Flavonols	103	25.55%
Sub Class: Flavanols	28	6.94%
Sub Class: Anthocyanins	16	3.97%
Sub Class: Isoflavonoids	5	1.24%
Class: Other polyphenols	23	5.69%
Sub Class: Hydroxybenzaldehydes	4	0.99%
Sub Class: Hydroxycoumarins	3	0.74%
Sub Class: Tyrosols	1	0.24%
Sub Class: Other polyphenols	15	3.72%
Class: Lignans. Sub Class: Lignans	9	2.23%
Class: Stilbenes. Sub Class: Stilbenes	3	0.74%

**Table 3 molecules-28-08133-t003:** Identified phenolic compounds in more than 5 species among the 24 species selected in this study. The percentage of representation in these species is also shown.

Phenolic Compound	Species in Which It Appears	Percentage (%)
Class: Phenolic acid. Sub Class: Hydroxybenzoic acids		
4-hydroxybenzoic acid	7	29.16%
Ellagic acid	8	33.33%
Gallic acid (3,4,5-trihydroxybenzoic acid)	17	70.83%
Hexahydroxydiphenoyl-glucose	6	25.00%
Protocatechuic acid (3,4-dihydroxy-benzoic acid)	9	37.50%
Syringic acid	8	33.33%
Vanillic acid	10	41.66%
Class: Phenolic acid. Sub Class: Hydroxycinnamic acids		
Caffeic acid (3,4-dihydroxycinnamic acid)	12	50.00%
Chlorogenic acid (3-*O*-caffeoylquinic acid)	13	54.16%
Cinnamic acid	7	29.16%
*p*-Coumaric acid	10	41.66%
Coumaroyl quinic acid	5	20.83%
Ferulic Acid	12	50.00%
Rosmarinic acid	5	20.83%
Methylrosmaric acid	6	25.00%
Class: Flavonoid. Sub Class: Flavanones		
Hesperetin 7-*O*-rutinoside (Hesperidin)	5	20.83%
Naringenin	8	33.33%
Naringenin 7-*O*-glucoside (Naringin-Prunina)	9	37.50%
Class: Flavonoid. Sub Class: Flavones		
Apigenin	11	45.83%
Apigenin 7-*O*-glucoside	6	25.00%
Luteolin (3′.4′.5.7-Tetrahydroxyflavone)	9	37.50%
Luteolin 7-*O*-glucoside	5	20.83%
Luteolin-7-*O*-rutinoside	6	25.00%
Class: Flavonoid. Sub Class: Flavonols		
Kaempferol	13	54.16%
Kaempferol *O*-hexoside	5	20.83%
Myricetin (3.3′.4′.5.5′.7-Hexahydroxyflavone)	9	37.50%
Myricetin 3-*O*-hexoside	6	25.00%
Myricetin-*O*-rhamnoside (Myricitrin)	12	50.00%
Myricetin-*O*-rutinoside	5	20.83%
Quercetin (3.3′.4′.5.7-Pentahydroxyflavone)	15	62.50%
Quercetin 3-*O*-glucoside (Isoquercetin)	16	66.66%
Quercetin-*O*-hexoside	8	33.33%
Quercetin-*O*-rhamnoside (Quercetrin)	11	45.83%
Quercetin 3-*O*-rutinoside (Rutin)	16	66.66%
Class: Flavonoid. Sub Class: Flavanols		
Catechin	11	45.83%
Epicatechin	10	41.66%
Epicatechin gallate	5	20.83%
Class: Flavonoid. Sub Class: Anthocyanins		
Cyanidin 3-*O*-glucoside	6	25.00%

**Table 4 molecules-28-08133-t004:** Species with the most represented phenolic compounds (present in over 10 of the 24 analyzed species).

					Phenolic Compounds										
	GA	VA	CA	CHA	*p*-CA	FA	Ap	K	MOR	Q	QOG	QOR	QORU	Ca	Epi
*C. ladanifer*	+	+				+	+		+	+	+	+	+	+	
*C. salviifolius*	+		+	+	+	+	+		+	+	+	+	+	+	+
*C. monspeliensis*	+						+		+	+	+	+	+		
*C. crispus*	+			+							+				
*C. albidus*	+			+				+	+	+		+	+		+
*C. populifolius*	+										+				
*C. multiflorus*	+	+	+	+	+	+	+			+	+		+		
*C. scoparius*	+		+	+			+	+		+	+		+		
*C. striatus*							+								
*E. multiflora*				+				+		+			+		
*E. scoparia*								+	+						
*E. australis*		+			+			+	+		+	+	+	+	+
*C. vulgaris*				+					+			+			
*M. communis*									+		+	+			
*P. lentiscus*	+	+	+	+	+	+		+	+	+	+	+	+	+	+
*P. terebinthus*	+		+		+	+		+	+	+		+	+	+	
*R.officinalis*	+	+	+		+	+	+	+		+			+		
*Q. ilex*	+	+	+			+			+	+	+		+	+	+
*Q. suber*	+					+								+	+
*A. unedo*	+	+	+	+	+	+		+	+		+	+	+	+	+
*L. stoechas*	+	+	+	+	+	+	+	+		+	+		+	+	+
*T. mastichina*			+	+			+	+		+	+				
*T. vulgaris*	+	+	+	+	+	+	+	+		+	+		+	+	+
*R. ulmifolius*	+	+	+	+	+	+	+	+		+	+	+	+	+	+

GA: Gallic acid; VA: Vallic acid; CA: Caffeic acid; CHA: Chlorogenic acid; *p*-CA: *p*-coumaric acid; FA: Ferulic acid; Ap: Apigenin; K: Kaemferol; MOR: Myricetin-O-rhamnoside; Q: Quercetin; QOG: Quercetin-O-glucoside; QORU: Quercetin-O-rutinoside; Ca: Catechin; Epi: Epicatechin.

## Data Availability

No new data were created or analyzed in this study. Data sharing is not applicable to this article.

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
