# Peer review of "Mediterranean Shrub Species as a Source of Biomolecules against Neurodegenerative Diseases"

_molecules, 2023, doi:10.3390/molecules28248133_

Round 1

Reviewer 1 Report

Comments and Suggestions for Authors

Abstract: In lines 11-12, the word ''treatments'' should be changed because it was used twice in the sentence, as well as in the previous sentence; in line 14, not only radicals generated in biological systems, it should be changed; ''source of resources'' should be reformulated; line 20, instead of several words ''acid'' should be one ''acids'';  p- should be in italics.

The authors should use the abbreviations AD and PD, in lines 67, 68, and 69; from the description of AD and PD pathogenesis it is not completely clear how antioxidant stress induces the diseases, thus it should be improved; line 81, these natural products should be mentioned and listed in the brackets; line 95, instead of -HO should be -OH; lines 1112-115, it is not clear where they act as inhibitors and agonists; line 125, synthesized resveratrol or isolated from plant material?; line 129-130, check typing mistakes (PDR); line 139, the full meaning of protein names should be given; line 158, some examples of the mentioned biotic and abiotic factors should be mentioned; line 188, phenolic acids and flavonoids are parts of polyphenols, why polyphenols were separated?; full Latin names should be given below Table 1; check missed bold letters; it is not clear how did the authors calculate the percentage in Table 2, thus it should be better explained; the presence of Figure 1 is completely unnecessary, particularly due to already mentioned reason - the difference in commercial interests and numbers of studies; line 228, the word ''families'' should be changed; the sentence in lines 232-235 should be reformulated and indicate that descending order is presented; the presence of the sentence in lines 237-239 is not clear when it is written in that way; m- and p- should be in italics; line 256, check typing mistakes () instead of []; line 236, via should be in italics; line 293, check typing mistakes; lines 295 and 299, FA instead of full meaning; line 338, check typing mistake (lowercase letter p-); line 342, LTP should be explained; -O-, in vitro and in vivo should be in italics; line 361, the compound is missed after ''and''; line 448, check typing mistakes.

Comments on the Quality of English Language

Minor editing is required.

Author Response

Dear Reviewer 1,

First of all we want to thank you for all the effort done in the corrections.

We think that the main criticisms about the manuscript are corrected now.

More specific points were addressed, which are commented below.

Abstract: In lines 11-12, the word ''treatments'' should be changed because it was used twice in the sentence, as well as in the previous sentence;

-Changed

in line 14, not only radicals generated in biological systems, it should be changed;

-Changed

''source of resources'' should be reformulated;  

-Changed

line 20, instead of several words ''acid'' should be one ''acids''; 

-Changed

p- should be in italics.

-Changed

The authors should use the abbreviations AD and PD, in lines 67, 68, and 69;

-Changed

From the description of AD and PD pathogenesis it is not completely clear how antioxidant stress induces the diseases, thus it should be improved;

- A brief description of how oxidative stress is related to disease has been added.

line 81, these natural products should be mentioned and listed in the brackets;

- Some examples have been added

line 95, instead of -HO should be -OH;

-Changed

lines 112-115, it is not clear where they act as inhibitors and agonists;

-This information has been specified

line 125, synthesized resveratrol or isolated from plant material?;

- In the cited reference it refers to resveratrol isolated from plants

line 129-130, check typing mistakes (PDR);

-Pdr5 changed to pdr5p

-PDR5 gene is correct

line 139, the full meaning of protein names should be given;

These proteins are mentioned as such in biochemical studies. Their names come from , Tor1(target of rapamycin), but there is no point in adding it. I have added the type of protein it is.

line 158, some examples of the mentioned biotic and abiotic factors should be mentioned;

(high temperatures and water stress in summer) already mentioned in the text, I have added herbivory.

line 188, phenolic acids and flavonoids are parts of polyphenols, why polyphenols were separated?;

-there was a mistake, it is “other polyphenols”

full Latin names should be given below Table 1;

Full latin names are mentioned under point 2: Description of the study area and article search strategy

-check missed bold letters;

-Ckecked,

It is not clear how did the authors calculate the percentage in Table 2, thus it should be better explained; the presence of Figure 1 is completely unnecessary, particularly due to already mentioned reason - the difference in commercial interests and numbers of studies;

-Figure 1 eliminated

- Sentence has been changed

line 228, the word ''families'' should be changed;

-Changed

the sentence in lines 232-235 should be reformulated and indicate that descending order is presented;

-Changed

the presence of the sentence in lines 237-239 is not clear when it is written in that way;

 m- and p- should be in italics;

-changed

line 256, check typing mistakes () instead of [];

-changed

line 236, via should be in italics;

-changed

line 293, check typing mistakes;

-changed

lines 295 and 299, FA instead of full meaning;

-changed

line 338, check typing mistake (lowercase letter p-);

-changed

line 342, LTP should be explained;

-Added

-O-, in vitro and in vivo should be in italics;

-changed

line 361, the compound is missed after ''and'';

-Added

line 448, check typing mistakes.

-Changed

We hope that the main points have been correctly addressed and the manuscript is now acceptable for publication.

Sincerely

N. Chaves

Reviewer 2 Report

Comments and Suggestions for Authors

This review article looks satisfactory overall, however the presentation of work and article writing needs improvement according to following suggestions.

1.      Revise the title of manuscript. A very superficial topic is selected for writing of this review.

2.      Revise the English language of whole manuscript with the help of some native English speaker/professional editing service, as the English language of this manuscript is weak.

3.      Rephrase the statement ‘providing an important source of resources’ in Abstract.

4.      Revise the statement ‘associated with the so-called “oxidative stress theory of ageing’ in line 33 in Introduction.

5.      Revise the lines 44-46 for clear understanding of meaning.

6.      Try to add schematic diagrams for major compounds mechanism of action in section 4 for easy understanding of readers.

7.      The manuscript is overloaded with repetition of similar information in many sections, please recheck whole manuscript and remove repetitive statements.

8.      It is better if the authors add 2 schematic figures depicting the specific mechanism of phenolics in treatment of neurodegeneration.

9.      Revise conclusion professionally. There are some unprofessional statements, which should be revised. 

Comments on the Quality of English Language

Please see attached review report

Author Response

Dear Reviewer 2,

First of all we want to thank you for all the effort done in the corrections.

We think that the main criticisms about the manuscript are corrected now.

More specific points were addressed, which are commented below.

This review article looks satisfactory overall, however the presentation of work and article writing needs improvement according to following suggestions.

  1. Revise the title of manuscript. A very superficial topic is selected for writing of this review.

- Title has been changed

  1. Revise the English language of whole manuscript with the help of some native English speaker/professional editing service, as the English language of this manuscript is weak.

The text has been revised

  1. Rephrase the statement ‘providing an important source of resources’ in Abstract.

-Changed

  1. Revise the statement ‘associated with the so-called “oxidative stress theory of ageing’ in line 33 in Introduction.

      -Revised

  1. Revise the lines 44-46 for clear understanding of meaning.

-Revised

  1. Try to add schematic diagrams for major compounds mechanism of action in section 4 for easy understanding of readers.

-Two figures have been added with diagrams showing the main neuroprotective mechanisms of phenolic acids and flavonoids.

  1. The manuscript is overloaded with repetition of similar information in many sections, please recheck whole manuscript and remove repetitive statements.

-Repetitive statements have been eliminated

  1. It is better if the authors add 2 schematic figures depicting the specific mechanism of phenolics in treatment of neurodegeneration.

-Two figures have been added with diagrams showing the main neuroprotective mechanisms of phenolic acids and flavonoids.

  1. Revise conclusion professionally. There are some unprofessional statements, which should be revised. 

- Conclusions have been changed

We hope that the main points have been correctly addressed and the manuscript is now acceptable for publication.

Sincerely

N. Chaves

Round 2

Reviewer 2 Report

Comments and Suggestions for Authors

The revised version of this review article looks satisfactory overall, however there are still some comments not addressed completely. The article writing needs improvement according to following suggestions.

1.      Revise the English language of whole manuscript with the help of some native English speaker/professional editing service, as the English language of this manuscript is still weak.

2.      Revise the statement ‘associated with the so-called “oxidative stress theory of ageing’ in line 33 in Introduction. Rephrase these lines. What is “so called”?

3.      Revise the lines 44-46 for clear understanding of meaning.

4.      Please revise the conclusion professionally. There are still some unprofessional statements, which are not well written and should be revised correctly.

Comments on the Quality of English Language

Please revise the manuscript with the help of some professional writing person of relevant field or native English speaker. 

Author Response

Dear Reviewer,

Thank you very much for your corrections and suggestions.

The suggested changes have been made and the English has been checked by an English professional.

The revised version of this review article looks satisfactory overall, however there are still some comments not addressed completely. The article writing needs improvement according to following suggestions.

  1. Revise the English language of whole manuscript with the help of some native English speaker/professional editing service, as the English language of this manuscript is still weak.

- English has been revised by an English professional.

  1. Revise the statement ‘associated with the so-called “oxidative stress theory of ageing’ in line 33 in Introduction. Rephrase these lines. What is “so called”?

-It has been changed

  1. Revise the lines 44-46 for clear understanding of meaning.

-It has been changed

  1. Please revise the conclusion professionally. There are still some unprofessional statements, which are not well written and should be revised correctly.

-It has been changed

We hope that the main points have been correctly addressed and the manuscript is now acceptable for publication.

Sincerely

N. Chaves
